# Learning Algorithmic Solutions to Symbolic Planning Tasks with a Neural Computer

## Abstract

A key feature of intelligent behavior is the ability to learn abstract strategies that transfer to unfamiliar problems. Therefore, we present a novel architecture, based on memory-augmented networks, that is inspired by the von Neumann and Harvard architectures of modern computers. This architecture enables the learning of abstract algorithmic solutions via Evolution Strategies in a reinforcement learning setting. Applied to Sokoban, sliding block puzzle and robotic manipulation tasks, we show that the architecture can learn algorithmic solutions with strong generalization and abstraction: scaling to arbitrary task configurations and complexities, and being independent of both the data representation and the task domain.

## 1 Introduction

Transferring solution strategies from one problem to another is a crucial ability for intelligent behavior (Silver et al., 2013). Current learning systems can learn a multitude of specialized tasks, but extracting the underlying structure of the solution for effective transfer is an open research problem (Taylor & Stone, 2009). Abstraction is key to enable these transfers (Tenenbaum et al., 2011) and the concept of *algorithms* in computer science is an ideal example for such transferable abstract strategies. An algorithm is a sequence of instructions, which solves a given problem when executed, independent of the specific instantiation of the problem. For example, consider the task of sorting a set of objects. The algorithmic solution, specified as the sequence of instructions, is able to sort any number of arbitrary classes of objects in any order, e.g., toys by color, waste by type, or numbers by value, by using *the same sequence of instructions*, as long as the features and compare operations defining the order are specified. Learning such structured, abstract strategies enables the transfer to new domains and representations (Tenenbaum et al., 2011). Moreover, abstract strategies as algorithms have built-in generalization capabilities to new task configurations and complexities.

Here, we present a novel architecture for learning abstract strategies in the form of algorithmic solutions. Based on the Differential Neural Computer (Graves et al., 2016) and inspired by the von Neumann and Harvard architectures of modern computers, the architectures modular structure allows for straightforward transfer by reusing learned modules instead of relearning, prior knowledge can be included, and the behavior of the modules can be examined and interpreted. Moreover, the individual modules of the architecture can be learned with different learning settings and strategies – or be hardcoded if applicable – allowing to split the overall task into easier subproblems, contrary to the end-to-end learning philosophy of most deep learning architectures. Building on memory-augmented neural networks (Graves et al., 2016; Neelakantan et al., 2016; Weston et al., 2015; Joulin & Mikolov, 2015), we propose a flexible architecture for learning abstract strategies as algorithmic solutions and show the learning and transferring of such in symbolic planning tasks.

### 1.1 The Problem of Learning Algorithmic Solutions

We investigate the problem of learning algorithmic solutions which are characterized by three requirements: **R1** – generalization to different and unseen task configurations and task complexities, **R2** – independence of the data representation, and **R3** – independence of the task domain.

Picking up the sorting algorithm example again, R1 represents the ability to sort lists of arbitrary length and initial order, while R2 and R3 represent the abstract nature of the solution. This abstraction enables the algorithm, for example, to sort a list of binary numbers while being trained only on hexadecimal numbers (R2). Furthermore, the algorithm trained on numbers is able to sort

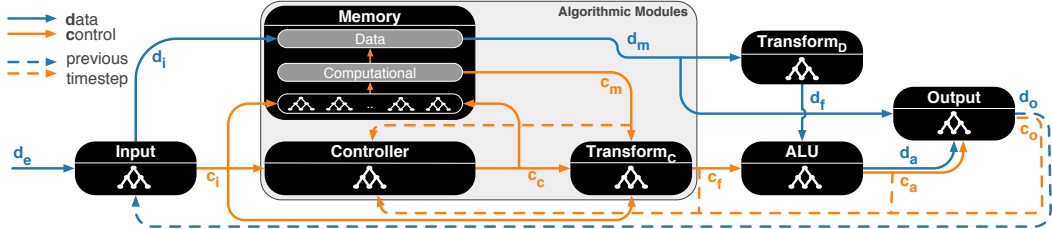

Figure 1: The proposed architecture with its modules inspired by computer architectures. In this work the modules are based on neural networks. Information flow is divided into *data* and *control* streams. The modules inside the highlighted area are learning the algorithmic solution in a reinforcement learning setting, whereas the others (data modules) are learned independently in a supervised setting or can use hardcoded information.

lists of strings (R3). If R1 – R3 are fulfilled, the algorithmic solution does not need to be retrained or adapted to solve unforeseen task instantiations – only the data specific operations need to be adjusted.

Research on learning algorithms typically focuses on identifying algorithmic generated patterns or solving *algorithmic problems* (Neelakantan et al., 2016; Zaremba & Sutskever, 2014; Kaiser & Sutskever, 2016; Kaiser & Bengio, 2016), less on finding *algorithmic solutions* (Joulin & Mikolov, 2015; Zaremba et al., 2016) fulfilling the three discussed requirements R1 – R3. While R1 is typically tackled, as it represents the overall goal of generalization in machine learning, the abstraction abilities from R2 and R3 are missing. Additionally, most algorithms require a form of feedback, using computed intermediate results from one computational step in subsequent steps, and a variable number of computational steps to solve a problem instance. Thus, it is necessary to be able to cope with varying numbers of steps and determining when to stop, in contrast to using a fixed number of steps (Neelakantan et al., 2016; Sukhbaatar et al., 2015), making the learning problem more challenging in addition.

A crucial feature for algorithms is the ability to save and retrieve data. Therefore, augmenting neural networks with different forms of external memory, e.g., matrices, stacks, tapes or grids, to increase their expressiveness and to separate computation from memory, especially in long time dependencies setups, is an active research direction (Graves et al., 2016; Weston et al., 2015; Joulin & Mikolov, 2015; Zaremba et al., 2016; Sukhbaatar et al., 2015; Kumar et al., 2016; Greve et al., 2016) with earlier work in the field of grammar learning (Das et al., 1992; Mozer & Das, 1993; Zeng et al., 1994). These memory-augmented networks improve performance on a variety of tasks like reasoning and inference in natural language (Graves et al., 2016; Weston et al., 2015; Sukhbaatar et al., 2015; Kumar et al., 2016), learning of simple algorithms and algorithmic patterns (Joulin & Mikolov, 2015; Zaremba et al., 2016; Graves et al., 2014), and navigation tasks (Wayne et al., 2018).

**The contribution** of this paper is a novel modular architecture building on a memory-augmented neural network (DNC (Graves et al., 2016)) for learning algorithmic solutions in a reinforcement learning setting. We show that the learned solutions fulfill all three requirements R1 – R3 for an algorithmic solution and the architecture can process a variable number of computational steps.

## 2   A NEURAL COMPUTER ARCHITECTURE FOR ALGORITHMIC SOLUTIONS

In this section, we introduce the novel modular architecture for learning algorithmic solutions, shown in Figure 1. The architecture builds on the Differential Neural Computer (DNC) (Graves et al., 2016) and its modular design is inspired by modern computer architectures, related to (Neelakantan et al., 2016; Weston et al., 2015).

The DNC augments a controller neural network with a differentiable autoassociative external memory to separate computation from memory, as memorization is usually done in the networks weights. The controller network learns to write and read information from that memory by emitting an interface vector which is mapped onto different vectors by linear transformations. These vectors control the read and write operations of the memory, called read and write heads. For writing and reading, multiple attention mechanisms are employed, including content lookup, temporal linkage and memory allocation. Due to the design of the interface and the attention mechanisms, the DNC is independent of the memory size and fully differentiable, allowing gradient-based end-to-end learning.

**Our architecture.** In order to learn algorithmic solutions, the computations need to be *decoupled* from the specific data and task. To enable such data and task independent computations, we propose multiple alterations and extensions to the DNC, inspired by modern computer architectures.

First, information flow is divided into two streams, data and control. This separation allows to disentangle data representation dependent manipulations from data independent algorithmic instructions. Due to this separation, the *algorithmic modules* need to be extended to include two memories, a data and a computational memory. The data memory stores and retrieves the data stream, whereas the computational memory works on information generated by the control signal flow through the learnable controller and memory transformations. The two memories are coupled, operating on the *same* locations, and these locations are determined by the computational memory, and hence by the control stream. As with the DNC, multiple read and write heads can be used. In our experiments, one read and two write heads are used, with one write head constrained to the previously read location.

In contrast to the DNC, but in line with the computer architecture-inspired design and the goal of learning deterministic algorithms, writing and reading uses hard attentions instead of soft attentions. Hard attention means that only one memory location can be written to and read from (unique addresses), instead of an weighted average over all locations as with soft attentions. Such hard attention was shown to be beneficial for generalization (Greve et al., 2016). We also employed an additional attention mechanism for reading, called *usage linkage*, similar to the temporal linkage of the DNC, but instead of capturing temporal relations, it captures *usage* relations, i.e., the relation between written memory location and previously read location. With both linkages in two directions and the content look up, the model has five attention mechanisms for reading. While the final read memory location is determined by a weighted combination of these attentions (see *attention* in Figure 5 in the Appendix), each attention mechanism itself uses hard decisions, returning only one memory location. See Appendix C for the effect of the introduced modifications and extensions.

For computing the actual solution, operating only on the control stream is not enough, as the model still needs to manipulate the data. Therefore, we added several modules operating on the data stream, inspired by the architecture of computers. In particular, an Input, Transform$_D$, ALU (*arithmetic logic unit*) and Output module were added (more details in Section 2.2). These modules manipulate the data, steered by the algorithmic modules. The full architecture is shown in Figure 1.

As algorithms typically involve recursive or iterative data manipulation, the model receives its own output as input in the next computation step, making the whole architecture an output-input model. With all aforementioned extensions, algorithmic solutions fulfilling R1 – R3 can be learned.

## 2.1 THE ALGORITHMIC MODULES

The algorithmic modules consist of the *C*ontroller, the *M*emory and the *T*ransform$_C$ module and build the core of the model. These modules learn the algorithmic solution operating on the control stream. With $t$ as the current computational step and $c$ as the control stream (see Figure 1), the input-output of the modules are $C(c_{i,t}, c_{m,t-1}, c_{f,t-1}, c_{a,t-1}, c_{o,t-1}) \longmapsto c_{c,t}$ , $M(c_{i,t}, c_{c,t}) \longmapsto c_{m,t}, d_{m,t}$ and $T_C(c_{c,t}, c_{m,t}, c_{i,t}) \longmapsto c_{f,t}$. The algorithmic modules are based on the DNC with the alterations and extensions described before. Next we discuss how these algorithmic modules can be learned before looking into the data-dependent modules.

### 2.1.1 LEARNING OF THE ALGORITHMIC MODULES

Learning the algorithmic modules, and hence the algorithmic solution, is done in a reinforcement learning setting using Natural Evolution Strategies (NES) (Wierstra et al., 2014). NES is a blackbox optimizer that does not require differentiable models, giving more freedom to the model design, e.g., the hard attention mechanisms are not differentiable. NES updates a search distribution of the parameters to be learned by following the natural gradient towards regions of higher fitness using a population of offsprings (altered parameters) for exploration. Let $\theta$ be the parameters to be learned and using an isotropic multivariate Gaussian search distribution with fixed variance $\sigma^2$, the stochastic natural gradient at iteration $t$ is given by

$$\nabla_{\theta_t} \mathbb{E}_{\epsilon \sim N(0,I)} \left[ u(\theta_t + \sigma \epsilon) \right] \approx \frac{1}{P\sigma} \sum_{i=1}^{P} u(\theta_t^i) \epsilon_i \ ,$$

where $P$ is the population size and $u(\cdot)$ is the rank transformed fitness (Wierstra et al., 2014). The parameters are updated by

$$\theta_{t+1} = \theta_t + \frac{\alpha}{P\sigma} \sum_{i=1}^{P} u(\theta_t^i)\epsilon_i \; ,$$

with learning rate $\alpha$. Recent research showed that NES and related approaches like Random Search (Mania et al., 2018) or NEAT (Stanley & Miikkulainen, 2002) are powerful alternatives in reinforcement learning. They are easier to implement and scale, perform better with sparse rewards and credit assignment over long time scales, have fewer hyperparameters (Salimans et al., 2017) and were used to train memory-augmented networks (Greve et al., 2016; Merrild et al., 2018).

For robustness and learning efficiency, weight decay for regularization (Krogh & Hertz, 1992) and automatic restarts of runs stuck in local optima are used as in (Wierstra et al., 2014). This restarting can be seen as another level of evolution, where some lineages die out. Another way of dealing with early converged or stuck lineages is to add intrinsic motivation signals like novelty, that help to get attracted by another local optima, as in NSRA-ES (Conti et al., 2018). In the experiments however, we found that within our setting, restarting – or having an additional *survival of the fittest* on the lineages – was more effective, see Appendix C for a comparison.

The algorithmic solutions are learned in a curriculum learning setup (Bengio et al., 2009) with sampling from old lessons (Zaremba & Sutskever, 2014) to prevent unlearning and to foster generalization. Furthermore, we created *bad memories*, a learning from mistakes strategy, similar to the idea of AdaBoost (Freund & Schapire, 1997), which samples previously failed tasks to encourage focusing on the hard tasks. This can also be seen as a form of experience replay (Mnih et al., 2015; Lin, 1992), but only using the task configurations, the initial input to the model, not the full generated sequence. Bad memories were developed for training the data-dependent modules to ensure their robustness and $100\%$ accuracy, which is crucial to learn algorithmic solutions. If the individual modules do not have $100\%$ accuracy, no stable algorithmic solution can be learned even if the algorithmic modules are doing the correct computations. For example, if one module has an accuracy of $99\%$, the $1\%$ error prevents learning an algorithmic solution that works *always*. This problem is even reinforced as the proposed model is an output-input architecture that works over multiple computation steps using its own output as the new input – meaning the overall accuracy drops to $36.6\%$ for 100 computation steps. Therefore using the bad memories strategy, and thus focusing on the mistakes, helps significantly in achieving robust results when learning the modules, enabling the learning of algorithmic solutions. While the bad memories strategy was crucial to achieve $100\%$ robustness when training the data-dependent modules, the effect on learning the algorithmic solutions was less significant (see Appendix C for an evaluation).

## 2.2 DATA-DEPENDENT MODULES

The data-dependent modules (Input, ALU, Transform$_\text{D}$ and Output) are responsible for all operations that involve direct data contact, such as receiving the input data from the *outside* or manipulating a data word with an operation chosen by the algorithmic modules. Thus, these modules need to be learned or designed for a specific data representation and task. However, as all modules only have to perform a certain subtask, these modules are typically easier to train.

As learning the algorithmic modules via NES does not rely on gradients and due to the information flow split, the data-dependent modules can be instantiated *arbitrarily*, e.g., can have non-differentiable parts, do not need to be neural networks or can be hardcoded. Therefore, prior knowledge can be incorporated by implementing it directly into these modules. The modular design facilitates the transfer of learned modules, e.g., using the same algorithmic solution in a new domain without retraining the algorithmic modules or learning a new algorithm within the same domain without retraining the data modules. Next the general functionality of the modules will be explained.

**The Input module** is the interface to the *external world* and responsible for data preprocessing. Therefore, it receives the external input data and the data from the previous computational step. It sends data to the memory and control signals to the subsequent modules with information about the presented data or the state of the algorithm – formally as $I(d_{e,t}, d_{o,t-1}) \longmapsto c_{i,t}, d_{i,t}$ .

**The ALU module** performs the basic operations which the architecture can use to modify data. Therefore, it receives the data and a control signal indicating which operation to apply and outputs

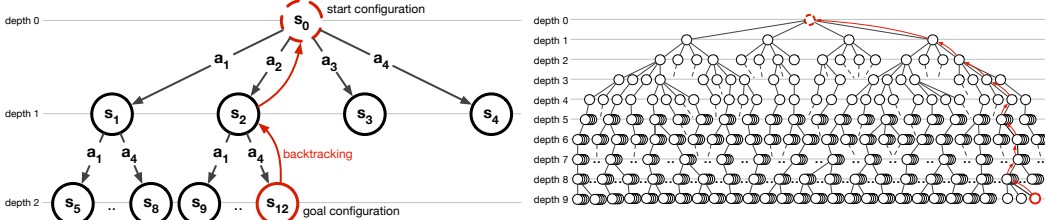

(a) Complexity that triggered learning.      (b) Complexity solved by the algorithmic solution.

Figure 2: Examples of search trees that the architecture implicitly learned to generate to solve a given symbolic planning task, where $s_i$ corresponds to task configurations and $a_k$ to ALU operations that transform the task configuration. **(a)** corresponds to a task from curriculum level 3 with a maximum number of computations steps of 15 including backtracking. **(b)** shows the tree for a task that required 330.631 computation steps (corresponds to level 82.656) that was solved by an algorithmic solution that triggered learning only until 15 steps, the complexity shown in (a).

the modified data and control signals about the operation – $A(c_{f,t}, d_{f,t}) \longmapsto c_{a,t}, d_{a,t}$. As in many applications the basic operations only modify a part of the data and to reduce the complexity of the ALU, a **Transform$_\mathbf{D}$ module** extracts the *relevant* part from the data beforehand – $T_D(d_{m,t}) \longmapsto d_{f,t}$ – or just transfers the unmodified data if no transformation is required for the task.

**The Output module** combines the result of the data manipulation operation from the ALU module and the data before the manipulation. It inserts the local change done by the ALU into the original data word – $O(c_{a,t}, d_{a,t}, d_{m,t}) \longmapsto c_{o,t}, d_{o,t}$. As before with the Transformation module, depending on the task, the Output module can also be designed to just pass on the received data.

## 3 EXPERIMENTS

We investigate the learning of symbolic planning tasks, where task complexity is measured as the number of computational steps required to solve a task, i.e., the size of the corresponding search tree (see Figure 2). Learning is done in the Sokoban domain, whereas the generalization and abstraction requirements R1 – R3 are shown by transferring to (1) longer planning tasks, (2) bigger Sokoban worlds, (3) a different data representation, and (4) two different task domains – sliding block puzzle and robotic manipulation.

In Sokoban, an agent interacts in a grid world with four actions – moving up, right, down or left. Therefore, the ALU can perform four operations and additionally a `nop` operation that leaves the given configuration unchanged. The world contains empty spaces that can be entered, walls that block movement and boxes that can be pushed onto empty space. A task is given by a start configuration of the world and the desired goal configuration. For learning, we use a world of size $6 \times 6$ that is enclosed by walls. A world is represented with binary vectors and four-dimensional one-hot encodings for each position, resulting in $144$-dimensional data words. The configuration of each world – inner walls, boxes and agent position – is sampled randomly. Each world is generated by sampling uniformly the number of additional inner walls from $[0, 2]$ and boxes from $[1, 5]$. The positions of these walls, boxes and the position of the agent are sampled uniformly from the empty spaces. An example task and the learned solution is shown in the Appendix in Figure 5 – the penguin is the agent, icebergs are boxes, iceblocks are walls and water is empty space.

### 3.1 ALGORITHMIC MODULES

In the experiments we use a feedforward neural network as Controller with a layer size of 16 neurons and `tanh` activation. The Transform$_\mathrm{C}$ is a linear layer projecting its 27-dimensional input onto the 5 operations of the ALU using `leaky-ReLU` activation and one-hot encoding. The computational memory has a word size of 8 bit, the Input module generates 3 control signals (2 for Learning to Search), and the ALU and Output module control signal feedback is not used here. Thus, the input to the Controller consists of 16 control signals and in total there are about 1600 parameters.

#### 3.1.1 LEARNING OF THE DATA-DEPENDENT MODULES

All data-dependent modules are trained in a supervised setting and consist of feedforward networks. They optimize a cross entropy loss using Adam (Kingma & Ba, 2015) on a mini-batch size of

20. To improve their generalization and robustness, the bad memories mechanism described in Section 2.1.1 is used with a buffer size of $200$ and $50\%$ of the samples within a mini-batch are sampled from that. The following task-dependent instantiations of the data-dependent modules are examples used for the Sokoban domain.

**The Input module** learns an equality function using differential rectifier units as inductive bias (Weyde & Kopparti, 2018) and consists of a feedforward network with 10 hidden units and `leaky-ReLU` activation. Using the learned binary equality signal $I_{e,t}$ at step $t$, it produces three binary control signals according to $c_{i,t}^{[1]} = (1 - I_{e,t}) - c_{i,t-1}^{[2]}$, $c_{i,t}^{[2]} = I_{e,t} + c_{i,t-1}^{[2]}$, and $c_{i,t}^{[3]} = I_{e,t} c_{i,t-1}^{[2]}$, indicating the different phases of the algorithm. For the Learning to Search experiment only the first two signals are used.

**The Transform$_{\mathbf{D}}$ module** extracts a different view on the data, if required by the ALU, as described in Section 2.2. Here, it consists of a feedforward network with 500 hidden neurons and uses `leaky-ReLU` activation. For the Sokoban domain, the actions that the agent can take – and therefore the operation the ALU can apply – only change the world locally. Thus, the Transform$_{\mathrm{D}}$ module extracts a local observation of the world $d_f$, i.e., the agent and the two adjacent locations in all four directions, as these are the only locations where an action can produce a change.

**The ALU module** receives the data view extracted by Transform$_{\mathrm{D}}$ and the control signal from Transform$_{\mathrm{C}}$, that encodes the operation to apply. It learns to apply the operations, i.e., it learns an action model by learning preconditions and effects, and outputs the (potential) local change together with a control signal indicating if the action changed the world or not. The local change is encoded as the direction of the change and the three according spaces. The module consists of two feedforward networks, one for the control signal $c_a$ and one for applying the actions producing the manipulated data $d_a$. The learned $c_a$ is used to gate the output between the output of the action network and the data input without change. The control network has two hidden layers with sizes $[64, 64]$, the action network has hidden layers with $[128, 64]$ neurons and both use `leaky-ReLU` activations.

**The Output module** inserts the (locally) changed data from the ALU into the data stream. It receives the data from the memory $d_m$, the data $d_a$ and control stream $c_a$ from the ALU. It consists of two feedforward networks for learning the data $d_o$ and the control signal $c_o$ stream. The control network has two hidden layers with sizes $[500, 250]$, the data network has hidden layers with $[500, 500]$ neurons and both use `leaky-ReLU` activations. The control signal $c_o$ is used for gating between the data with the inserted change and the original data $d_m$. To ensure that the Output module uses the manipulated data of the ALU and is not learning to manipulate the data itself, it is constrained to learn a binary mask that indicates where the change needs to be inserted. This binary map indicates for each position in $d_a$ where to insert it in $d_m$ and can be seen as a structured prediction problem. Note, the training data only consists of data and control signals, the true binary mask is not known.

### 3.2 LEARNING ALGORITHMIC SOLUTIONS

We investigate the learning of two algorithms, (1) a search algorithm and (2) a search-based planning algorithm. The data-dependent modules do not need to be retrained for the different algorithms. For evaluating that the learned strategy is an abstract *algorithmic solution*, we show that it fulfills the three requirements R1 – R3 discussed in Section 1.1.

#### 3.2.1 LEARNING TO SEARCH

In the first task, the model has to learn breadth-first-search to find the desired goal configuration. For that purpose, the initial input to the model is the start and goal configuration and subsequent inputs are the goal configuration and the output of the model from the previous computation step. To solve the task, the model has to learn to produce the correct search tree and recognizing that the goal configuration is reached by choosing the `nop` operation for the correct computation step.

For the curriculum learning the levels are defined as the number of nodes from the search tree that have to be fully explored, e.g., for Level 1, up to five correct computation steps have to be performed on the initial configuration; for Level 3 the initial configuration as well as the two subsequently found configurations need to be fully explored (see Figure 2(a)). This requires up to 13 correct computational steps. Curriculum levels are specified up to Level 21 that involves up to 85 correct computation steps to be solved. An additional Level 22 is activated afterwards that consists of new samples from all 21 levels for evaluation. To prevent unlearning of previous levels, $20\%$ of the samples in the mini-batch are sampled uniformly from previous levels. As in (Wierstra et al., 2014)

we use restarting, but here the run automatically restarts if the maximum fitness of a level is not reached within 2500 iterations. All experiments have a total budget of 10.000 iterations.

The fitness function $f$ uses step-wise binary losses computed as comparison to the correct solution over mini-batches of $N$ samples and is defined as

$$f = \begin{cases} \frac{1}{N}\sum_n^N f_e^{[n]} & \text{if } \frac{1}{N}\sum_n^N f_e^{[n]} < 100 \\ \frac{1}{N}\sum_n^N f_e^{[n]} + f_b^{[n]} & \text{otherwise} \end{cases} \text{, with} \tag{1}$$

$$f_e^{[n]} = \frac{100}{3T_e^{[n]}}\sum_{t=1}^{T_e^{[n]}} I(c_{f,t}^{[n]} = \tilde{c}_{f,t}^{[n]}) + 2I(d_{m,t}^{[n]} = \tilde{d}_{m,t}^{[n]}) \quad \text{and} \quad f_b^{[n]} = 20I(c_{f,T_e^{[n]}+1}^{[n]} = \texttt{nop}),$$

where $T_e$ is the number of steps required for constructing the search tree or when the first mistake occurs, $c_{f,t}$ is the operation chosen to be applied by the ALU from Transform$_C$ at step $t$, $d_{m,t}$ is the data word read from the memory, and $\tilde{c}_{f,t}$ and $\tilde{d}_{m,t}$ are the correct choices respectively. The exploration fitness $f_e^{[n]}$ captures the fraction of correct computation steps until the goal configuration is found, scaled to 0-100%. Note that, NES therefore only uses a single scalar value that summarizes the performance of the parameters over $N$ samples and all computational steps. The learning rate $\alpha$ is to 0.01, the $\sigma$ of the search distribution to 0.1, weight decay is applied with 0.9995, mini-batch size is $N = 20$ and the population size is $P = 20$.

We use a gini coefficient based ranking that gives more importance to samples with higher fitness (Schaul et al., 2010). The maximum fitness is 120 for all levels and a level is solved when 250 subsequent iterations have the maximum fitness, i.e., 5000 samples are solved correctly. The bad memories consist of 200 samples and 25% of the samples within a mini-batch are sampled uniformly from those. Whenever 10 subsequent iterations achieve the maximum fitness, the buffer is cleared and *no learning is performed*.

### 3.2.2 LEARNING TO PLAN (SEARCH + BACKTRACK)

In the second task, the model has to learn, in addition to the breadth-first-algorithm that computes a search tree to the goal configuration, to also extract the path from the search tree that encodes the solution to the given planning problem (see Figure 2 and Figure 5 in the Appendix). Therefore, the model has to not only learn to encode and perform two different algorithms, but also to switch between them at the correct computation step.

The initial input to the model is the start and goal configuration and subsequent inputs are the goal configuration and the output of the model from the previous computation step, as before. When the goal configuration is found by the model, the input is the start configuration and the previous output. To solve the task, the model has to learn to produce the search tree and recognizing that the goal configuration is reached as before. In addition, after recognizing the goal configuration, the model needs to switch behavior and output the path of the search tree encoding the planning solution. This solution consists of the states from the initial to the goal configuration and `nop` operations in reverse order. Therefore, the number of maximum computation steps increases up to 89 in Level 21. The fitness function is defined as in Equation equation 1 but with

$$f_b^{[n]} = \frac{50}{3T_b^{[n]}}\sum_{t=T_e^{[n]}+1}^{T_e^{[n]}+T_b^{[n]}} I(c_{f,t}^{[n]} = \texttt{nop}) + 2I(d_{m,t}^{[n]} = \tilde{d}_{m,t}^{[n]}),$$

where $T_b$ is the number of steps required for backtracking the solution or when the first mistakes occurs. The maximum fitness is 150 and all other settings remain as before.

### 3.3 **R1** – GENERALIZATION TO UNSEEN TASK CONFIGURATIONS AND COMPLEXITIES

A main goal in all learning tasks, is to achieve generalization – to not only learn to solve seen situations, but to learn a solution that generalizes to unseen situations. One evaluation of this generalization ability is built into our learning process itself. A curriculum level is solved after 250 subsequent iterations (5000 samples) with maximum fitness and iterations with maximum fitness do not trigger learning. Thus, if presenting a new level that involves more complex tasks, the fitness stays at maximum and no learning is triggered, the previously learned solution generalizes to the new setting – generalizes to more complex tasks (see Figure 2).

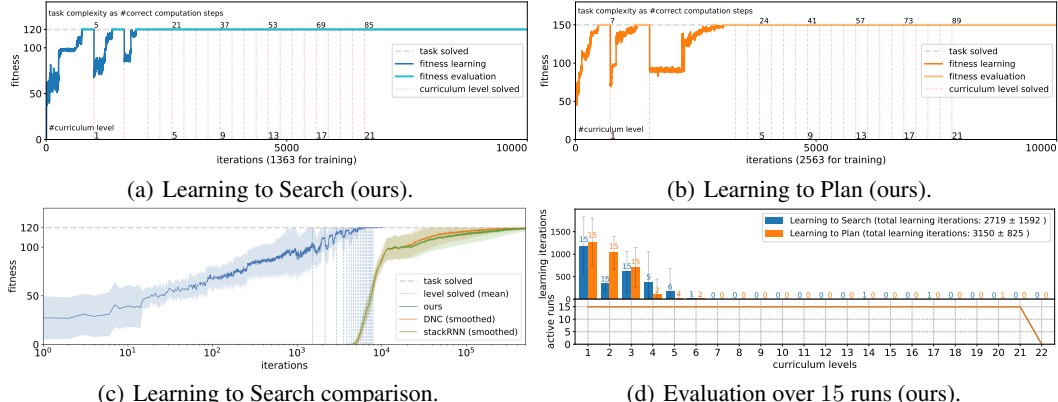

(a) Learning to Search (ours).  (b) Learning to Plan (ours).

(c) Learning to Search comparison.  (d) Evaluation over 15 runs (ours).

Figure 3: **(a-c)** The gray dashed line marks the maximum fitness and the colored lines show the fitness. The light colors indicate that the maximum fitness is achieved and no learning is triggered. The colored dashed lines indicate when a curriculum level was solved successfully. When no learning is triggered after a new level is unlocked, the model generalized to more complex tasks. The top numbers indicate the number of computational steps the model needs to perform correctly to solve samples of the associated curriculum level. **(c)** Comparison with the original DNC and a stack-augmented neural network on the *Learning to Search* task over 10 runs. In contrast to our architecture, both methods are trained in a supervised setup with gradient descent and cross-entropy loss, i.e., have a richer and localized training signal. For comparison the mean and standard deviation of the same fitness function that our model uses for training is shown. Both are not able to successfully solve Level 1 within considerably more iterations. **(d)** 15 runs of the two learning tasks, highlighting that learning happens during the first levels and generalizes to the subsequent levels. Bar plot shows mean and standard deviation of the number of learning iterations, numbers on top of the bars show the number of runs that triggered learning in that level. Lower plot shows the number of runs that solved the according curriculum level, i.e., where they ended after the budget of 10.000 iterations.

This generalization is shown in Figure 3. For example, in the *Learning to Plan* setup (Figure 3(b)), after 3 levels the algorithmic solution is found and no learning is triggered anymore during the run. Moreover, the last triggered learning was for curriculum Level 3 – meaning a complexity of 15 computational steps – and the found solution generalizes up to the highest specified curriculum Level 21 with 89 computational steps. Learning the algorithmic solution is done within 3 levels and 2563 iterations. Figure 3(d) shows the evaluation of learning to solve the two tasks over 15 runs each. In contrast, the original DNC (Graves et al., 2016) model and a stack-augmented recurrent neural network for algorithmic patterns (Joulin & Mikolov, 2015) are not able to solve Level 1 when trained in a supervised setup with gradient descent and considerably more training iterations, see Figure 3(c) and the Appendix B for implementation details.

**Task complexity.** Additionally, we evaluated the learned algorithmic solution with task complexities far beyond the specified curriculum learning levels, i.e., complexities experienced during training. Therefore, we used the run shown in Figure 3(b) and solved tasks requiring **330.631** computational steps (corresponds to level 82.656), having been **trained only up to 15** steps (see Figure 2 for the complexities) and having been tested during training only up to 89 steps. Remember the models recurrent output-input structure, given the initial task input, the model performs 330.631 computational steps, i.e., learns to build a search tree with over 330.600 nodes, autonomously correct to compute and output the solution. Moreover, the solution learned in $6 \times 6$ environments, successfully solved all tasks within $8 \times 8$ environments. Thus, the learned strategy represents an abstract algorithmic solution that generalizes and scales to arbitrary task configurations and complexities, fulfilling R1. The learned algorithmic solution is explained with an example in the Appendix A.

## 3.4  **R2** – INDEPENDENCE OF THE DATA REPRESENTATION

Algorithmic solutions are independent of the data representation, meaning the abstract strategy is still working if the encoding is changed, as long as the data-dependent operations are adjusted. Consider again a sorting algorithm. Its algorithmic behavior stays the same independent of if it has to sort a list of numbers encoded binary or hexadecimally, as long as the compare operators are defined. To show that our learned algorithmic solutions have this feature and fulfill R2, we change the representation of the data, but reuse the learned algorithmic modules and the model can

Figure 4: Transferring the learned algorithmic solution *(left)* to a new data representation (R2) and *(middle & right)* to two new task domains (R3). In all setups, all 200.000 samples over all curriculum levels are solved correctly without triggering learning, indicated by the constant maximum fitness, showing the straightforward transfer of the learned solution – the abstract features R2 and R3 of the learned solution.

still solve all tasks without retraining. The data-dependent modules are adapted and relearned. The changed representation, e.g., the penguin represents a wall instead of the agent, and results over 10.000 iterations (200.000 samples) over all curriculum levels are shown in Figure 4 (*left*). The fitness is at maximum from the start, showing that all samples in all levels are successfully solved without triggering learning while operating on the new data representation and hence, R2 is fulfilled.

### 3.5   **R3** – INDEPENDENCE OF THE TASK DOMAIN

Requirement R3 states that an algorithmic solution is independent of the task domain. Consider again the sorting algorithm example: as long as the compare operators are defined, it is able sort *arbitrary objects*. Therefore, the data-dependent modules are adapted and relearned but we reuse the learned algorithmic solution on two new task domains.

As new domains, $3 \times 3$ *sliding block puzzles* and a *robotic manipulation* task are used (Figure 4). Configurations are represented with binary vectors as described for Sokoban in Section 3. For the puzzle domain, actions are sliding adjacent tiles onto the free (white) space from four directions. A task configuration is given as a start and goal board configuration. In the robotic manipulation domain, a task is given as start and goal configuration of the objects. The available actions are the four locations on which objects can be stacked, e.g., the action `pos1` encodes to move the gripper to the position and place the grasped object on top, or to pick up the top object if no object is grasped. The maximum stacking height is 3 boxes, resulting in a discrete representation of the object configuration with a $3 \times 4$ grid. As with the new data representation, the learned algorithmic solution is able to solve all 200.000 presented samples from all curriculum levels in the new domains without triggering learning (Figure 4), showing the independence of the task domain, fulfilling R3.

## 4   CONCLUSION

We present a novel architecture for learning algorithmic solutions and showed how it can learn abstract strategies that generalize and scale to arbitrary task configurations and complexities (R1) (Section 3.3), and are independent to both, the data representation (R2) (Section 3.4) and the task domain (R3) (Section 3.5). Such algorithmic solutions represent abstract strategies that can be transferred directly to novel problem instantiations, a crucial ability for intelligent behavior.

To show that our architecture is capable of learning strategies fulfilling the algorithm requirements R1 – R3 in symbolic planning tasks, we performed experiments with complexities orders of magnitude higher than seen during training (15 vs. 330.631 steps, and Figure 2 & 3), and transferred the learned solution to bigger state spaces, a new data representation and two new task domains (Figure 4) – showing, to the best of our knowledge, for the first time how such abstract strategies can be represented and learned with memory-augmented networks. The learned algorithmic solution can be applied to any problem that can be framed as such a symbolic search or planning problem.

The modular structure and the information flow of the architecture enable the learning of algorithmic solutions, the transfer of those, and the incorporation of prior knowledge. Using Natural Evolution Strategies for learning removes constraints on the individual modules, allowing for arbitrary module instantiations and combinations, and the beneficial use of a non-differentiable memory module (Greve et al., 2016). As the complexity and structure of the algorithmic modules need to be specified, it is an interesting road for future work to learn these in addition, building on the ideas from Greve et al. (2016); Merrild et al. (2018). Showing how algorithmic solutions characterized by R1 – R3 can be represented and learned with memory-augmented networks sets the foundation for future work, extending beyond symbolic planning and incorporating intrinsic motivation (Oudeyer & Kaplan, 2009; Baldassarre & Mirolli, 2013) to discover new and unexpected strategies.

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

## A    BEHAVIOR OF THE LEARNED ALGORITHMIC SOLUTION

Figure 5 highlights the learned algorithmic behavior – one memory location is read with content lookup attention repeatedly until all operations have been applied, the node is fully explored. Then attention shifts towards temporal linkage to read the *next* data to be explored. This pattern continuous until the goal configuration is found in step 11. After that, behavior changes to output the backtracking solution by switching to usage linkage attention and `nop` operations until reaching the initial configuration.

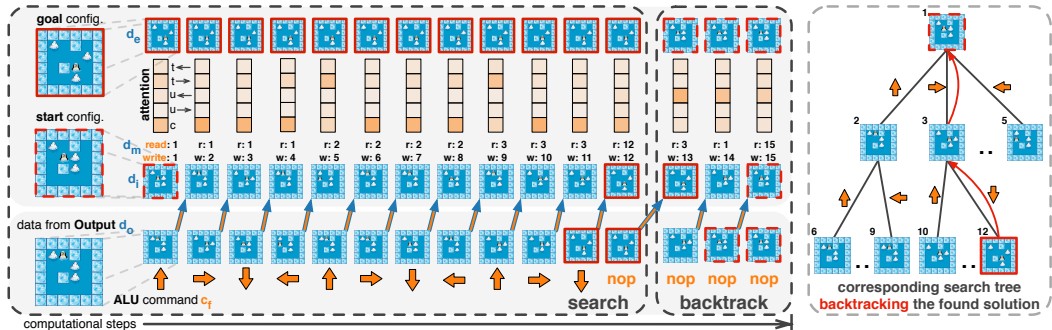

Figure 5: The behavior of the learned model on a task from Level 3 (see Sec. 3.2 for details) and the corresponding search tree that is constructing implicitly. In the *search* phase, the model fully explores one node by successively applying all operations, before reading the next node, until the goal is found. Then behavior changes in the *backtrack* phase, where the solution of the planning task is emitted as the states from start to goal in reverse order along with `nop` operations. The algorithmic behavior can also be seen in the repetitive patterns of the attention vector, showing the five attention mechanisms for reading (temporal and usage linkage in both directions, and content lookup), that represents how strong each mechanism for reading is used in each computation step.

## B    DETAILS ON THE IMPLEMENTATIONS OF THE COMPARISON METHODS

Both models, the orignal Differential Neural Computer (DNC) (Graves et al., 2016) and the stack-augmented recurrent network (Joulin & Mikolov, 2015) are trained in a supervised setting with cross-entropy losses for 500.000 iterations to compensate the pretraining of the data modules. They use the same output-input loop as our architecture, i.e., receiving their own output as input in the next computation step in addition to the goal configuration. The loss is computed based on the correct sequences of configurations and the control signal indicating that the goal has been reached, similar like the fitness function from our architecture in equation 1. Both use a LSTM network with 256 hidden units as controller and the memory word size is set to 152, equal to our model. Like our architecture, the DNC has one read and two write heads. The stack-augmented model uses four stacks with the three actions PUSH, POP, and NO_OP.

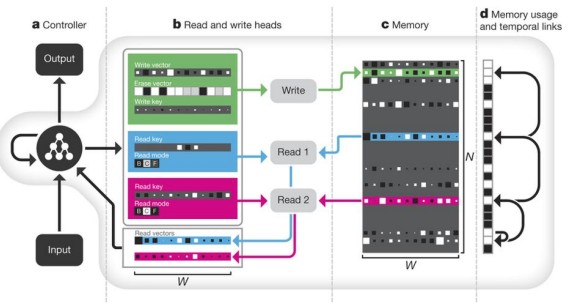
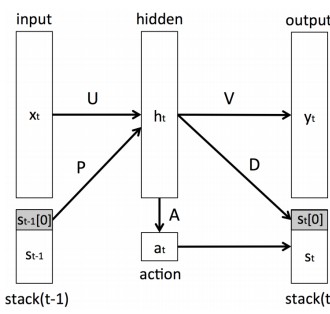

(a) Differential Neural Computer, reprinted with permission from (Graves et al., 2016).

(b) Stack-augmented recurrent network, reprinted with permission from (Joulin & Mikolov, 2015) .

Figure 6

## C EVALUATION OF THE LEARNING PROCESS AND MODEL COMPONENTS

For evaluation the effect of the individual modifications and extensions we compared our architecture with and without them on the Learning to Search task. In all setups all runs had a budget of 10.000 iterations. The bar plots show mean and standard deviation of the number of learning iterations, numbers on top of the bars show the number of runs that triggered learning in that level. Plots below the bar plot show the number of runs that successfully solved the according curriculum level, i.e., where they ended after the budget of 10.000 iterations. All comparisons are done without the restarting mechanisms, except in the evaluation for that mechanism.

### NOVELTY AND RESTARTS

Here two mechanisms to face the problem of getting stuck in local optima are evaluated, namely the automatic restart as in the original NES (Wierstra et al., 2014) and the use of an additional novelty signal as in NSRA-ES (Conti et al., 2018). For the novelty calculation, we defined the behavior as the sequence of read memory locations and applied ALU operations. The baseline model does not use either of the two mechanisms. While we did not observe an improvement using novelty, the automatic restarts reduced the number of learning iterations, see Figure 7. Note that the baseline and novelty model are also able to learn algorithmic solutions, but they require more iterations and, hence, they die out before the final curriculum level due to reaching the budget of 10.000 iterations.

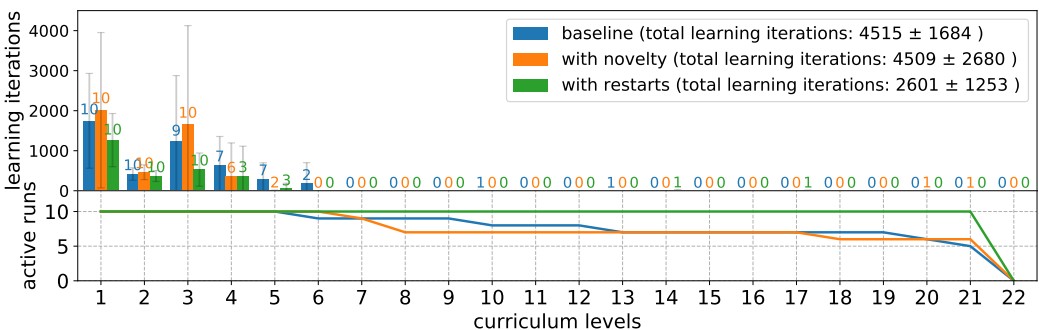

Figure 7: Evaluation of an additional novelty signal and automatic restarts.

### CONSTRAINED WRITE HEAD

Here we evaluated the introduced constrained write head, that updates the previously read memory location. We compared against two models without this constrained head, one with one write head and one with two write heads to compensate the missing constrained head. The constrained head was a necessary modification to enable the efficient learning of algorithmic solutions, see Figure 8.

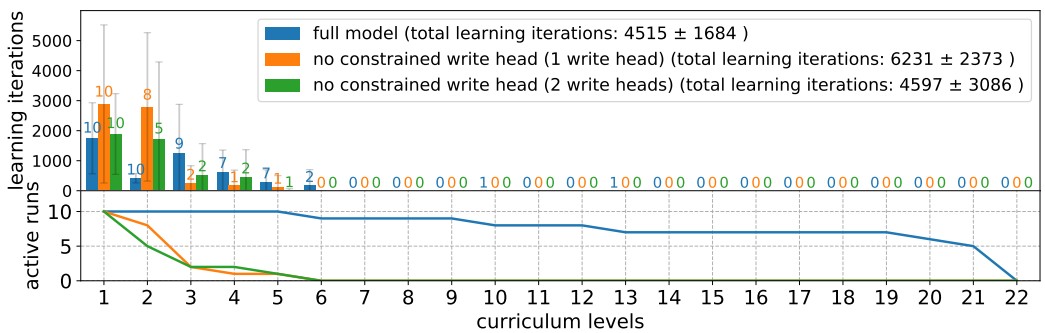

Figure 8: Evaluation of the introduced constrained write head.

USAGE-LINKAGE AND HARD ATTENTION VS. SOFT ATTENTION

Here the introduced usage-linkage and hard attention mechanism for memory access are evaluated. While using hard attention instead of soft attention was a necessary modification to enable efficient learning of algorithmic solutions, the introduced usage-linkage had a smaller impact on the Learning to Search task, as shown in Figure 9. When applied to the Learning to Plan setup however, the usage-linkage improved the learning of algorithmic solutions significantly, see Figure 10. Both results show that the model learns to use the attention mechanisms that are required for the algorithmic solution, i.e., the usage-linkage is especially useful for the backtracking in the Learning to Plan setup compared to the Learning to Search setup where no backtracking is required.

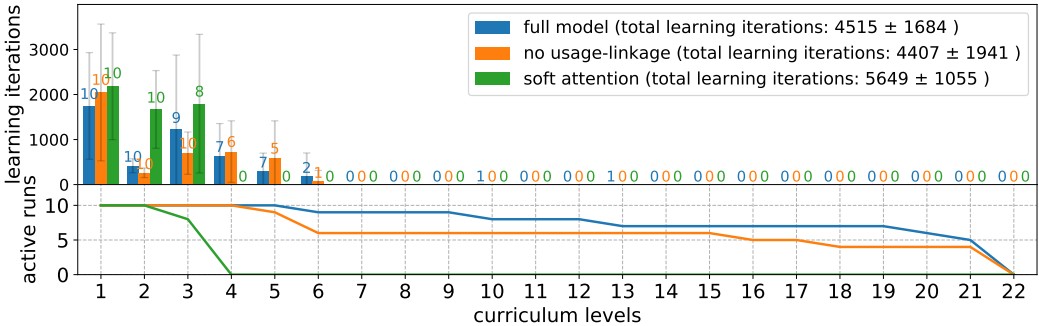

Figure 9: Evaluation of the introduced usage-linkage attention and the hard attention memory access.

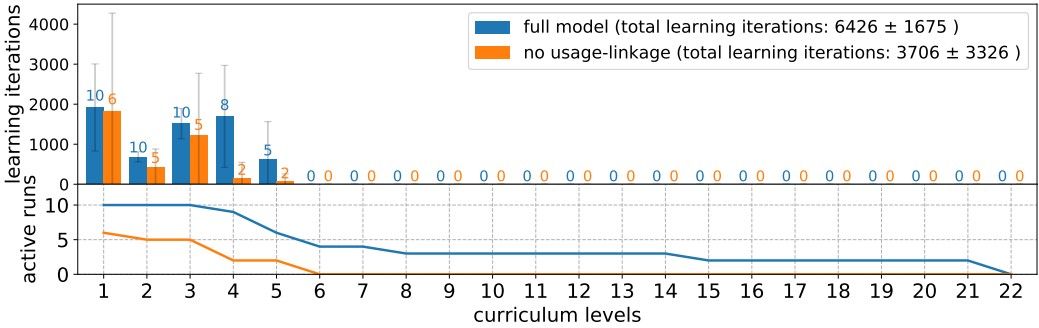

Figure 10: Evaluation of the usage-linkage attention on the Learning to Plan setup.

BAD MEMORIES

The bad memories approach was developed while learning the data-dependent modules and was a necessary mechanism to learn robust and generalized modules with 100% accuracy, as explained in Section 2.1.1. For learning the algorithmic solutions, the impact of this learning from mistakes strategy was less significant, see Figure 11.

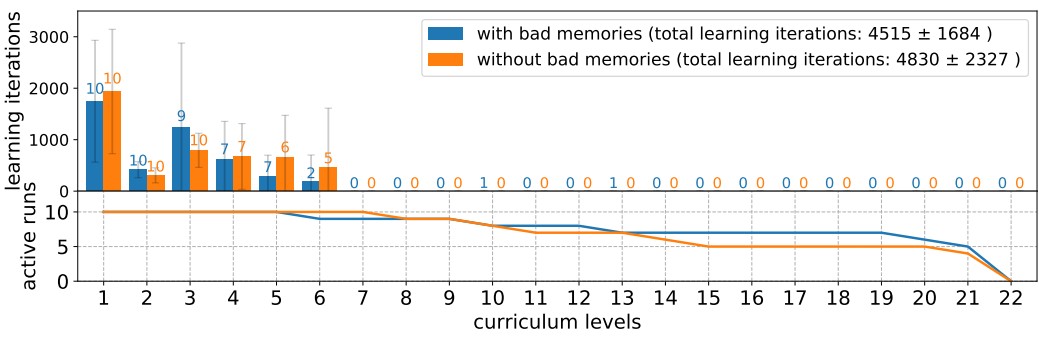

Figure 11: Evaluation of the bad memories mechanism.

