# OpenReview forum: "Learning Algorithmic Solutions to Symbolic Planning Tasks with a Neural Computer"
_ICLR.cc/2020/Conference — Reject_

### Official Review · AnonReviewer1 · 2019-10-18
**Official Blind Review #1**

**Rating:** 3

**Review:**

This paper proposes modifications and modular extensions to the differential neural computer (DNC). The approach is nicely modular, decoupling the data modules from algorithmic modules. This enables the authors to pretrain the data modules with supervised learning and to train the small algorithmic modules with neural evolution strategies (NES). NES is a global optimization method (which may be understood as policy gradients where the parameters of the neural policy are the actions) and consequently this enables the authors to use discrete selection mechanisms instead of the soft attention mechanisms of DNC.

The authors correctly argue for generalization to different and unseen tasks (R1), independence of data representation (R2) and independence of task environment (R3). These are important points, and the authors are able to achieve these by using a modular approach where the interface modules are pretrained.

While what can be achieved with modularity is important, the modules themselves are rather simple and leave open the question of scalability. NES clearly does not scale to higher dimensional spaces, and the complexity or real say vision modules is the problem for many people --- one does not just learn the vision module and freeze it.

The idea that modularity can be used to attain greater generality and domain independence has already been explored at ICLR to some extent. In fact, some authors have shown theoretically provable generalisation via verification approaches: https://openreview.net/forum?id=BkbY4psgg

In summary, this paper demonstrated the advantages of modular neural systems, but fails to address the important issue of making sure the modules scale to real problems.




**Experience Assessment:**

I have published in this field for several years.

**Review Assessment: Checking Correctness Of Derivations And Theory:**

I assessed the sensibility of the derivations and theory.

**Review Assessment: Checking Correctness Of Experiments:**

I assessed the sensibility of the experiments.

**Review Assessment: Thoroughness In Paper Reading:**

I read the paper at least twice and used my best judgement in assessing the paper.

---

> ### Author Response · Authors · 2019-11-11
> **We thank you for your feedback and helpful comments.**
>
> “NES clearly does not scale to higher dimensional spaces”
> Population based blackbox optimization is currently challenging ‘traditional’ gradient based training for deep learning, and was shown to scale to millions of parameters [1-4].
> Furthermore, the modular design of the architecture enables the use of smaller models, and the complexity of the data modules is not affected by that.
>
> [1] Conti et al., Improving Exploration in Evolution Strategies for Deep Reinforcement Learning via a Population of Novelty-Seeking Agents, NeurIPS 2018
> [2] Chrabaszcz et al., Back to Basics: Benchmarking Canonical Evolution Strategies for Playing Atari, IJCAI 2018
> [3] Such et al., Deep Neuroevolution: Genetic Algorithms are a Competitive Alternative for
> Training Deep Neural Networks for Reinforcement Learning, arXiv 2017
> [4] Salimans et al., Evolution strategies as a scalable alternative to reinforcement learning. arXiv 2017
>
> “one does not just learn the vision module and freeze it.”
> The problem of learning robust and adaptive vision modules is an open question itself, that is beyond the scope of this paper. Here, we focus on learning the algorithmic modules, i.e., the algorithmic solution, focusing on achieving generalization and abstraction while assuming the data modules are working correctly.
>
> “In fact, some authors have shown theoretically provable generalisation via verification approaches: https://openreview.net/forum?id=BkbY4psgg”
> While this paper indeed shows the benefits of recursive solutions for generalization of neural programming architectures, it is learned from program traces that already include the recursive structure. In contrast, our architecture is learned from the data sequences the algorithm produces, and discovers the recursive solution from that (see Figure 5 for the repetitive pattern of the learned solution). Additionally, using NES and a single scalar value to grade the all computational steps and all samples in a minibatch, leads to sparse and challenging learning setup. Moreover and in contrast, we show the direct transfer of the learned solutions to novel task representations and task domains.
>
> “In summary, this paper demonstrated the advantages of modular neural systems, but fails to address the important issue of making sure the modules scale to real problems.”
> Beside the advantages of modular systems, we show that abstract strategies in form of algorithmic solutions fulfilling the three specified requirements (R1-R3), can be represented and learned from data sequences instead of program traces in a general memory-augmented network based architecture. This fundamental research enables further investigation on more and challenging problems. Additionally, the solution learned in a gridworld computer game is transferred amongst others to a robotic manipulation task, in simulation and on the real robot.

---

### Official Review · AnonReviewer2 · 2019-10-23
**Official Blind Review #2**

**Rating:** 6

**Review:**

This paper introduces a neural controller architecture for learning abstract algorithmic solutions to search and planning problems. By combining abstract and domain-specific components, the model is able to mimic two classical algorithms quite closely across several domains. The precision of the learning is very high; verified by generalization to substantially larger problem sizes and different domains. One notable conclusion is that Evolutionary Strategies is able to learn algorithmic solutions whose precision is on par with deterministic algorithms. The method of triggering learning based on curriculum level performance is a notable feature that nicely couples generalization progress with learning, and yields insightful learning curves.

Although the paper shows that learning abstract algorithmic solutions is possible, it is not clear that the framework is able to learn such solutions when the training signals have not already been broken down into their smallest constituent parts. In other words, it is not clear what this framework could be used for, since it appears the experimenter must already possess a complete specification of the target algorithm and breakdown of the domain. Is there a setting where this framework could be used to learn something the experimenter is not already aware of? Or is the main point that it is technically possible to get an NN to learn this behavior?

Although it is clear that models are achieved that satisfy R1-R3, it is not clear exactly what problem formulation is being considered. It would be very helpful if the paper included a formal problem definition so that the purpose of each framework component and differences w.r.t. prior work are clear.

Similarly, the motivation for each of the data dependent modules is not clear. Is there something fundamental about this particular decomposition into modules? Or are these just the modules that were necessary given the specifics of the algorithms that were learned in experiments? How does this framework generalize to other kinds of algorithms?

Are the comparison methods (DNC and stackRNN) unable to generalize to larger problem sizes? Including the full comparisons on generalization would give a more complete picture. Similarly, the figures are missing the comparisons for Learning to Plan for DNC and stackRNN.

Is the comparison w.r.t. training time in Figure 3c fair, since the proposed framework pretrains the submodules?

Is there a fundamental problem of DNC being addressed here? E.g., are there some critical types of submodules where making them differentiable is not an option?

Is the algorithm limited to cases where the number of actions at each state is equal? I.e., could it be applied to algorithms like shortest path in the DNC paper?

Finally, the last line talks about intriguing applications to the real world, but the running example in the paper is sorting. Is there some hypothetical but concrete example of how this framework could help in the real world, and do something better than a hard-coded classical algorithm? Or discover a new algorithm?

----------

After the rebuttal and discussion, I am convinced the paper is a useful contribution, and have increased my rating. However, I still think the presentation can be much improved to improve the clarity of the contribution. I would hope to see the following addressed in the final version:

1. More detailed and precise discussion of how the approach relates to prior work. As is, this discussion is informal and scattered throughout the paper. Enumerating the distinctions in one place would make the contribution much more clear.

2. The above would also help make the explicit contribution more clear. E.g., the "Contribution" paragraph currently does not contain the fact that the machine has only been applied to "planning" problems. This should be included to avoid making the contribution seem overly general.

3. More formal description of what each module does. Right now, they are described informally. Actually seeing the equations of what each computes would make it easier to understand how and why they all fit together.

I still see the main contribution as "It is technically possible to train the abstract controller of a neural computer for planning using NES, so that R1-R3 are satisfied." Ideally, the above clarifications would make it more clear that this is the main contribution, or that there are some other key contributions beyond this.


**Experience Assessment:**

I have read many papers in this area.

**Review Assessment: Checking Correctness Of Derivations And Theory:**

I assessed the sensibility of the derivations and theory.

**Review Assessment: Checking Correctness Of Experiments:**

I assessed the sensibility of the experiments.

**Review Assessment: Thoroughness In Paper Reading:**

I read the paper at least twice and used my best judgement in assessing the paper.

---

> ### Author Response · Authors · 2019-11-11
> **We thank you for your nice summary and  detailed comments.**
>
> “.. could be used to learn something the experimenter is not already aware of? Or is the main point that it is technically possible to get an NN to learn this behavior?”
> The main point is to show that such abstract strategies in form of algorithmic solutions fulfilling the three specified requirements (R1-R3) can be represented and learned from data sequences instead of program traces in a general memory-augmented network based architecture, and that these abstract solutions can be transferred directly to novel representations and domains. While it is correct that we use the data sequences of the target algorithms (instead of program traces as widely done) for learning, it is only used to calculate one single scalar value over all computational steps and samples in the mini-batch. By using a different strategy for calculating the score that does not use the data sequences but, e.g., intrinsic motivation signals, the architecture could potentially discover other solutions. This is part of future research, building on this work here, that has shown the general capability of the architecture first.
>
> “It would be very helpful if the paper included a formal problem definition so that the purpose of each framework component ..”
> The purpose of the individual modules is described in general and formally in Section 2.1 for the algorithmic modules and in Section 2.2 for the data dependent modules. In Section 3.1.1 the instantiations of the data  modules for the sokoban domain are explained.
>
> “Is there something fundamental about this particular decomposition into modules? Or are these just the modules that were necessary given the specifics of the algorithms that were learned in experiments? How does this framework generalize to other kinds of algorithms?”
> The main idea behind this decomposition comes from the design of modern computer architectures and due to the split into data and control information flow. The framework was designed with these principles and to be generous. For different problem instances, the data modules can be instantiated as necessary, or not used at all by just implementing an identity function, as described in Section 2.2. The application to other kinds of algorithms is part of ongoing research, building on the first successful learning and representing of algorithmic solutions fulfilling R1-R3 presented here.
>
> “Are the comparison methods (DNC and stackRNN) unable to generalize to larger problem sizes? .. Similarly, the figures are missing the comparisons for Learning to Plan for DNC and stackRNN.”
> As shown in Figure 3c, the comparison methods did not solve curriculum level 1 and therefore failed to generalize here. Learning to Plan is missing for these methods, as they failed on the Learning to Search task, which is a subtask of the Learning to Plan.
>
> “Is the comparison w.r.t. training time in Figure 3c fair, since the proposed framework pretrains the submodules?”
> The comparisons methods were trained up to 500.000 iterations to compensate the pretraining of the data modules (they were trained for 300.000 iterations). As shown in Figure 3c, even with this budget and a richer learning signal (they were trained in a supervised fashion with ground truth feedback on every computational step), they did not solve level 1 successfully, while our architecture generalized to all complexities in about 3000 iterations.
>
> “..problem of DNC being addressed here? E.g., are there some critical types of submodules where making them differentiable is not an option?”
> The main ‘problem’ of the DNC to fulfill R1-R3 is the missing separation between data and control information, i.e., between data and algorithmic modules. For efficient learning and representing such deterministic behavior, hard attention was a crucial change that is not differentiable, please see Figure 9 for that comparison.
>
> “.. limited to cases where the number of actions at each state is equal? I.e., could it be applied to algorithms like shortest path in the DNC paper?”
> The Input module could output the available actions per state such that the algorithmic modules could learn to apply these actions. As the algorithmic solution learned by our architecture is performing breadth-first-search, it does find the shortest path (assuming equal edge costs). In contrast to the DNC paper, the graph is not presented to the model, but rather it has to build it itself, and the DNC papers shortests paths were of length 5, whereas our solution solved shortest paths up to 9, see Figure 2 and Section 3.1.
>
> “.. and do something better than a hard-coded classical algorithm? Or discover a new algorithm?”
> Discover a new algorithm is a very interesting point, indeed. As mentioned before, by using, for example, an intrinsic motivation based calculation of the fitness function, the architecture may explore unknown solutions to the given problem. This is part of future research that can now be started building on the presented foundation.

---

### Official Review · AnonReviewer3 · 2019-10-24
**Official Blind Review #3**

**Rating:** 6

**Review:**

This paper presents an approach called a neural computer, which has a Differential Neural Computer (DNC) at its core that is optimised with an evolutionary strategy. In addition to the typical DNC architecture, the system proposed in this paper has different modules that transfer different domain representations into the same representation, which allows the system to generalise to different and unseen tasks.

The paper is interesting and well written but I found that the contributions of this paper could be made more clear.

First, the idea of evolving a Neural Turing machine was first proposed in Greve et al. 2016, which the authors cite, but only in passing in the conclusion. Greve et al. paper introduced the idea of hard attention mechanisms in an NTM through evolution and the benefits of having a memory structure that does not have to be differentiable. However, if the reader of this paper is not careful, they would miss this fact. I, therefore, suggest featuring Greve’s paper more prominently and highlighting the differences/similarities to the current paper earlier in the introduction.

Second, the idea of learned modules to allow the approach to work across different domains is interesting, but I’m wondering how novel it really is? Isn’t this basically just like feature engineering and changing the underlying representation, something that we have been doing for a long time? Also, the domains that this approach can be applied to seem potentially limited in that the two problems have to already be very similar; In fact, I probably wouldn’t call them different tasks but the same task with a different visual representation.

I also had a question about the DNC training. Is the DNC version also trained with NES? It would be good to know how much of the difference between the proposed approach and the DNC is because of the training method (NES vs SGD) or other factors.

Once the points raised above and the specific contributions of this paper are made more clear, I would suggest accepting it.

####After rebuttal####
The authors' response and the revised paper address most of my concerns now. Since I do believe the approach could be more novel, I'm keeping my weak accept, but do think it is very interesting work.

**Experience Assessment:**

I have published one or two papers in this area.

**Review Assessment: Checking Correctness Of Derivations And Theory:**

I assessed the sensibility of the derivations and theory.

**Review Assessment: Checking Correctness Of Experiments:**

I assessed the sensibility of the experiments.

**Review Assessment: Thoroughness In Paper Reading:**

I read the paper thoroughly.

---

> ### Author Response · Authors · 2019-11-11
> **We thank you for your valuable feedback.**
>
> The contribution of this paper is a novel modular architecture building on a memory-augmented neural network that can represent and learn algorithmic solutions in a reinforcement learning setting using NES. The learned solutions fulfill the three specified requirements for algorithmic solutions: R1 – generalization to different and unseen task configurations and task complexities, R2 – independence of the data representation, and R3 – independence of the task domain. By fulfilling these requirements, the solutions learned by our architecture generalize to complexities far beyond the ones encounter during training, and can be transferred directly to other representations and domains.
>
> “First, the idea of evolving a Neural Turing machine was first proposed in Greve et al. 2016, which the authors cite, but only in passing in the conclusion. .. “
> Thank you for pointing out the missing hard attention mechanism reference in the beginning, we will update the paper accordingly. Beside the hard attention, Greve et al. use evolution strategies to evolve the structure of the NTM. In contrast, our focus is to show how abstract strategies fulfilling R1-R3 can be represented and learned in a memory-augmented based neural architecture. To incorporate the approach of evolving the structure, or parts of it, in addition, is an interesting next step and therefore discussed in the conclusion.
>
> “Second, the idea of learned modules to allow the approach to work across different domains is interesting, but I’m wondering how novel it really is? Isn’t this basically just like feature engineering and changing the underlying representation, something that we have been doing for a long time?”
> The presented architecture can be seen as a combination of symbolic manipulation and deep learning, whereas the data modules provide the symbols on which the algorithmic modules learn the abstract strategies.
>
> “Also, the domains that this approach can be applied to seem potentially limited in that the two problems have to already be very similar”
> The learned algorithmic solution is building trees/graphs and using backtracking to extract the planning solution. Thus, any problem that can be framed as a search or symbolic planning problem can be solved by the learned solution.
>
> “I also had a question about the DNC training. Is the DNC version also trained with NES?”
> No, the DNC and the stackRNN are trained in a supervised fashion with gradient descent (see Figure 3c caption, Section 3.3 and Appendix B), and thus, they use a richer learning signal, giving localized feedback in every computational step. In contrast, using NES on our architecture, only a single scalar value is used to score all computational steps and all samples in the mini-batch.

---

> > ### Comment · AnonReviewer3 · 2019-11-12
> > **More clarification needed**
> >
> > Thank you for clarifying but I feel like my questions could have been addressed in more detail.
> >
> > "Thank you for pointing out the missing hard attention mechanism reference in the beginning, we will update the paper accordingly. Beside the hard attention, Greve et al. use evolution strategies to evolve the structure of the NTM" -> This paper also showed the benefits of having a memory that is non-differentiable and using evolution in general to optimize the weights (not just structure) of a NTM; it is not only the hard attention mechanism that is related to this work. I hope the authors can update the paper before the discussion deadline to reflect this.
> >
> > "The presented architecture can be seen as a combination of symbolic manipulation and deep learning, whereas the data modules provide the symbols on which the algorithmic modules learn the abstract strategies" -> This response does not really answer the question of "Isn’t this basically just like feature engineering and changing the underlying representation, something that we have been doing for a long time?"
> >
> > "The learned algorithmic solution is building trees/graphs and using backtracking to extract the planning solution. Thus, any problem that can be framed as a search or symbolic planning problem can be solved by the learned solution. " -> what problems would this exclude and is this aspect highlighted in the paper?

---

> > > ### Author Response · Authors · 2019-11-12
> > > **Thank you for the discussion.**
> > >
> > > “This paper also showed the benefits of having a memory that is non-differentiable and using evolution in general to optimize the weights (not just structure) of a NTM; it is not only the hard attention mechanism that is related to this work. I hope the authors can update the paper before the discussion deadline to reflect this.“
> > > We agree that in addition to the non-differentiable memory that is due to the hard attention mechanism, using an evolution strategy to learn the model is another connection. We apologize for not clarifying this in our answer. Both links will be updated in the paper.
> > >
> > > “This response does not really answer the question of "Isn’t this basically just like feature engineering and changing the underlying representation, something that we have been doing for a long time?"”
> > > The focus of the paper is to show how to represent and learn algorithmic solutions fulfilling the R1-R3. Especially for enabling the abstract features R2 and R3, the information flow has to be divided into only data relevant and control/algorithmic signals. The learned mapping to the control signals can be seen as extracting abstract features for the algorithmic solution, we agree. Nevertheless, to the best of our knowledge this has not been implemented in an unified architecture and been used in the context of learning such abstract strategies fulfilling all three requirements. To represent and learn solutions on this abstraction level, such a mapping is crucial, whether it is learned or hardcoded.
> > >
> > > “"The learned algorithmic solution is building trees/graphs and using backtracking to extract the planning solution. Thus, any problem that can be framed as a search or symbolic planning problem can be solved by the learned solution. " -> what problems would this exclude and is this aspect highlighted in the paper? “
> > > In this paper, we focused on symbolic planning tasks. As long as the problem can be framed as such, e.g., having the symbols and their manipulations, and having an initial and desired goal state, the solution can be applied. But to clarify, this applies to the learned solution, not the architecture itself. Using a different learning problem, a different fitness calculation, other problems than such symbolic planning can be investigated. Again, here we focus on such problems as they are not trivial to be learned (see comparisons) and to show the benefit of investigating algorithmic solutions with R1-R3. We’ll clarify this in the paper.

---

### Decision · Program_Chairs · 2019-12-19

**Decision:**

Reject

**Comment:**

The authors present a method that optimizes a differentiable neural computer with evolutionary search, and which can transfer abstract strategies to novel problems.  The reviewers all agreed that the approach is interesting, though were concerned about the magnitude of the contribution / novelty compared to existing work, clarity of contributions, impact of pretraining, and simplicity of examples.  While the reviewers felt that the authors resolved the many of their concerns in the rebuttal, there was remaining concern about the significance of the contribution.  Thus, I recommend this paper for rejection at this time.